# 3D Inverse modeling of EM-LIN data for the exploration of coastal sinkholes in Quintana Roo Mexico

Marco A. Perez-Flores[2], Luis E. Ochoa-Tinajero[1] and Almendra Villela y Mendoza[3]

[1]Posgrado en Ciencias de la Tierra, CICESE, Mexico.
[2]Centro de Investigación Científica y de Educación Superior de Ensenada (CICESE), Mexico.
[3]Universidad Autónoma de Baja California (UABC), Mexico.

*Correspondence to*: Marco Pérez-Flores (mperez@cicese.mx)

**Abstract.** In the Yucatan Peninsula (YP), southern Mexico, cities and towns are settled on a platform of calcareous sedimentary sequence, where karst processes have formed numerous sinkholes, underground water conduits, and caverns. Anthropogenic activities there threaten the only source of freshwater supply, which is in a regional unconfined aquifer; there are no lakes or rivers on the surface. For the sustainable management of this resource in the YP, mathematical tools are needed in order to model groundwater. To determine the geometry of the aquifer, for example the positions of caves, sinkholes, and underground principal conduits, we modified a software to invert three-dimensional electromagnetic low-induction number (3D EM-LIN) data for a set of profiles at arbitrary angles. In this study we used the EM-LIN geophysical method to explore the Chac-Mool sinkhole system in the state of Quintana Roo, Mexico. We performed inverse modeling in 3D using the EM-34 instrument for vertical and horizontal magnetic dipoles. The 3D inversion process yields models that enable us to correlate the path of the underground principal conduits with the subsurface electrical resistivity. In this work we show that inverse modeling of EM-LIN data can give us information about how close to surface are the underground water conduits and the location of the boundary between fresh and salty water.

## 1 Introduction

The main source of fresh water in the Yucatan Peninsula is a regional unconfined karst aquifer that is constituted by sedimentary limestones (Bauer-Gottwein et al., 2011). Karst aquifers are extremely vulnerable to contaminants because of their high permeability and the peculiar turbulent groundwater flow passing through karst conduits and caves (Worthington et al., 2001; Parise et al., 2015; Parise, 2019). Rapid population growth and coastal tourism in the state threaten the only source of freshwater supply in the peninsula.

In order to guarantee the sustainable use of this groundwater resource knowledge on the hydrogeological characteristics, such as geometry and position, of caverns and sinkholes and the depth of the freshwater/saltwater mixing zone (halocline), three-dimensional data inversion is needed.

Sinkholes are natural geological features connecting the land surface with underground karst terrains, and they are formed when rainwater dissolves limestone, creating underground voids (Coskun, 2012). Two main groups of sinkholes have been

identified in the genetic classification (Williams, 2004; Gutierrez et al., 2014). The first group comprises solution sinkholes, which are formed by differential corrosion, lowering the ground surface where karst rocks are exposed. The second group comprises subsidence sinkholes, which result from both subsurface dissolution and downward gravitational movement.

In Quintana Roo many sinkholes, caverns and underground water conduits have been reported by scuba divers, and the Quintana Roo Speleological Survey has produced an underground map of the Riviera Maya for tourism purposes. However, geophysical techniques have rarely been applied as noninvasive approaches to explore this area (Estrada-Medina et al., 2010; Gondwe et al., 2010; Beauer-Gottwein et al., 2011). Electrical resistivity tomography has proven effective for exploring karst areas (Ahmed and Carpenter 2003, Chalikakis et al. 2011); however, in the Quintana Roo region the lack of soil on the hard limestone terrain has made placing electrodes a complicated and time-consuming task, raising expenses for data collection. New approaches to geophysical and coastal karst prospecting are therefore needed to develop and maintain sustainability plans in the YP.

In this study we aim to explore a novel approach by using electromagnetic (EM) methods at low-induction numbers (LIN) and applying 3D geophysical inverse modeling (Perez-Flores et al., 2012) in order to set up a conceptual model of a sinkhole system and gain more knowledge on the geomorphology of the site. The methodology and results could be useful tools for the management of the Quintana Roo coastal zones, which is important for tourism and requires accurate information for prospect plans of development.

We did not find references on the use of EM-LIN in karst systems, but we found that the Direct Current (DC) and aero-TDEM (Time Domain Electromagnetic Method) were used for the Sian-Kan natural reserve by Supper et al. (2009). These authors performed EM-34 measurements but they did not do any further processing, like performing geophysical inversion.

## 1.1 Study area

2    This research was carried out in the Yucatan Peninsula (YP), an area largely dominated by karst landscape (Bauer-Gottwein et al., 2011). From the geological point of view, the YP is constituted by a sequence of calcareous sediments (Bonet and Butterlin, 1962) and is characterized by its flat landform (no topography) and the absence of surface rivers. A review of the YP karst aquifer is well described by Bauer-Gottwein et al. (2011), and an extended description of coastal cave development is given by Smart et al. (2006).

Our study area covers the Chac-Mool sinkhole and is 20 km south of Playa del Carmen in the state of Quintana Roo (approximately 20°30'46.37" N and 87°14'49.32" W) (Fig. 1). The area extends to 1 km² and is fully covered by dense vegetation. The ground presents high secondary porosity. Annual precipitation there is around 1,200 mm and topography is a flat surface with a slope of 9 m above sea level within 20 km from the shoreline (CNA, 2016). The hydraulic gradient in the southern part of Playa del Carmen was estimated at 58-130 mm/km (Beddows, 2004). Due to its proximity to the coast (2 km), the study area is penetrated by seawater. Water intrusion is dependent on tides and rainfall (Beddows, 2004). Chac-Mool is a sinkhole complex where two underground water conduits presumably connect the Little-Brother sinkhole and the Air-Dome sinkhole. The underground river pathways in some sections have been documented on maps made by

scuba divers (Quintana Roo Speleological Survey) but other sections and vertical components remain unknown. The entire rock matrix is possibly saturated with fresh and brackish water in pores and small conduits. The apparent conductivity is high because it averages the matrix conductivity (low value) with the seawater conductivity (high value).

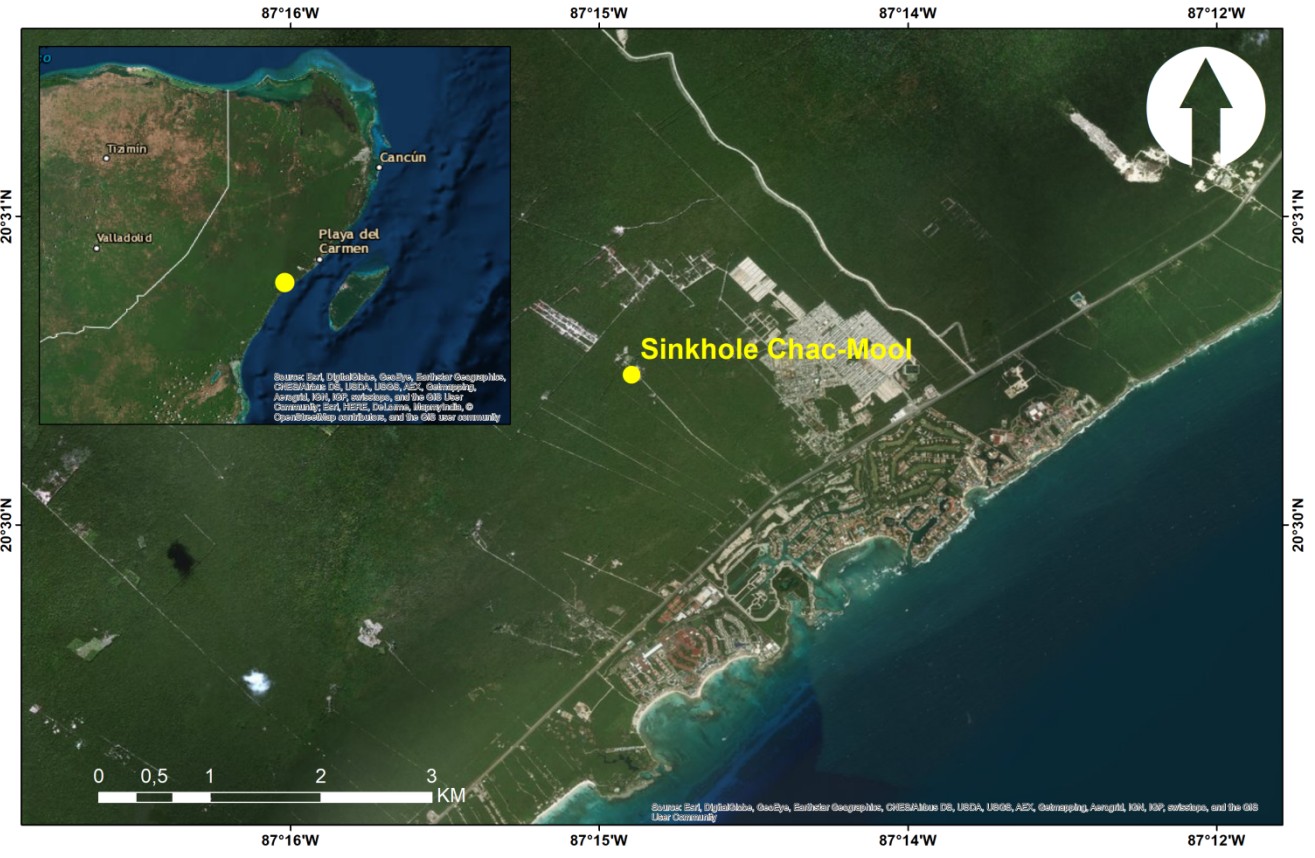

Figure 1. Study area: Chac-Mool sinkhole in the state of Quintana Roo, Mexico.

## 1.2 Electromagnetic survey

In September 2015, we carried out a field survey over the study area. We obtained seven profiles with the EM-34 (Geonics) instrument, which operates within the LIN domain as described in McNeill (1980). The main reason for using the EM-34 is that it can accurately obtain data in a more easy and faster way in terrains with no soil, expediting field work in hard terrains. The basic principle consists in the transmission of an alternating current of constant frequency ($f$) through a coil, which generates a primary electromagnetic field ($\boldsymbol{H_p}$) that induces electrical currents in the conductive bodies embedded in the subsoil (following Faraday's Law). A secondary electromagnetic field in the ground ($\boldsymbol{H_s}$) is then generated by the conductive bodies. These two fields differ in amplitude and phase, and they are detected by a coil (receiver) that is separated by a distance $s$(m)

from the transmitter. The induction number, $N$, is defined as the quotient between $s$(m) and the skin depth $\delta$(m): $N = s(m)/\delta(m)$. At LIN ($N<1$) the imaginary part of $\boldsymbol{H_s/H_p}$ is a straight line for which the slope is the conductivity of a homogeneous half-space. Because the ground is not homogenous, we say we get an apparent conductivity: $\sigma_a = (4/\omega\mu_0 S^2)(H_s/H_p)$.

Both loops (source and receiver) are commonly used on the same plane (coplanar). We have two possible arrays, one when both loops are parallel to the earth's surface (vertical magnetic dipoles or VMD) and the other when both loops are perpendicular to the earth's surface (horizontal magnetic dipoles or HMD). The separation between loops can be extended to 10 m, 20 m, and 40 m in both arrays. For this study, measurements were made along 6 lines (Fig. 2), and the observation points were spaced every 5 m. Because vegetation in the jungle was dense, we were unable to locate profiles anywhere, and so we

took the paths around the Chack-Mool, Little Brother, and Air Dome sinkholes. The distribution of the six profiles covers approximately a rectangular area. Therefore, we performed a 3D inversion in order to follow the three-dimensional complexity of the sinkholes. For the 3D inverse modeling we followed the method by Perez-Flores et al. (2012) but the algorithm they used was designed for profiles that were measured in parallel ($0^0$) or perpendicular ($90^0$) positions with respect to the other profiles. Later on, we show how we modified the equations for arbitrary angle profiles. The length of the six profiles (1 to 6)

varies between 50 m and 140 m (Fig. 2).

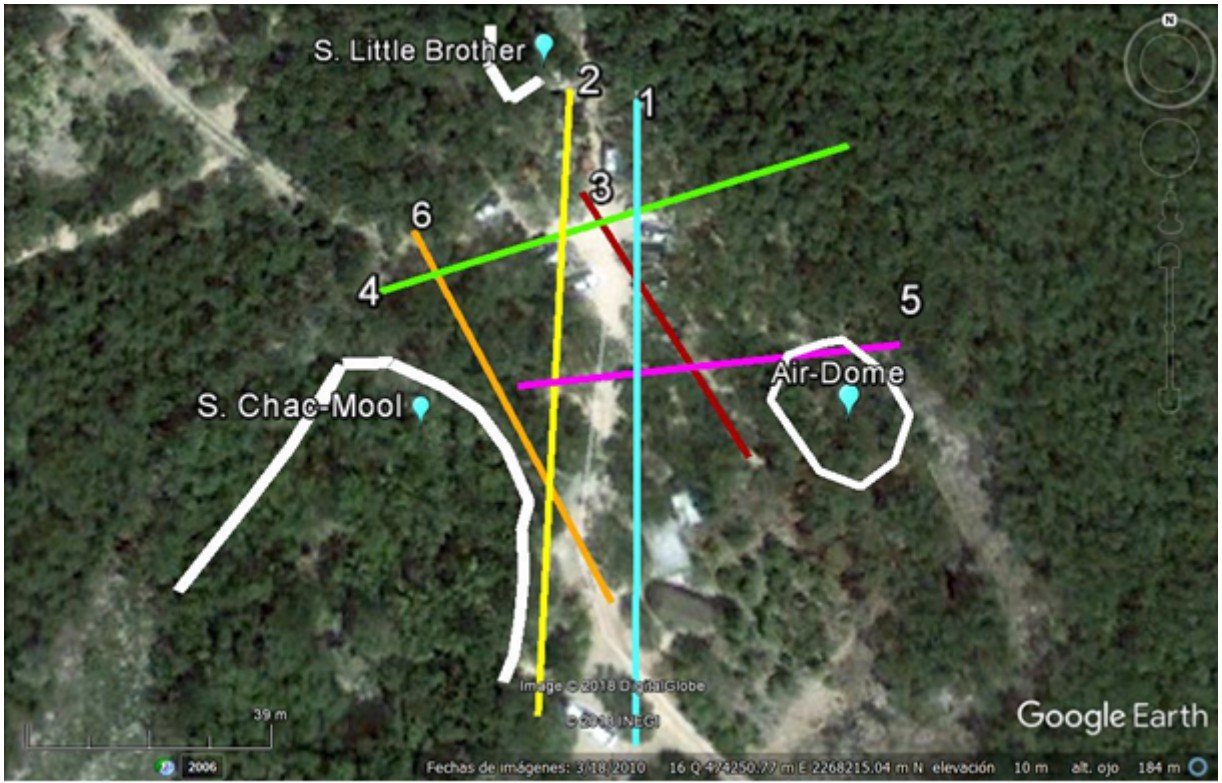

Figure 2. EM survey on the Chac-Mool sinkhole. The numbered profiles cross the hidden underground water conduits. White lines mark the sinkholes.

## 1.3 Inverse modelling

EM data (apparent conductivity, $\sigma_a$) were assumed to be approximately the weighed average of the subsurface electrical conductivity distribution, as described by Pérez-Flores et al. (2004). We associated the apparent conductivity ($\sigma_a$) with the true subsurface conductivity ($\sigma$) by means of a weighting function (that is, the Green function and electric-field product) using the integral equation formulated by Pérez-Flores et al. (2001):

$$\sigma_a(\boldsymbol{r_2}, \boldsymbol{r_1}) \cong -\frac{16\pi s}{\omega \mu_0 m} \int_v \boldsymbol{G}(\boldsymbol{r_2}, \boldsymbol{r}) \cdot \boldsymbol{E}(\boldsymbol{r}, \boldsymbol{r_1}) \sigma(\boldsymbol{r}) dv \tag{1}$$

where $\boldsymbol{r_1}$ and $\boldsymbol{r_2}$ are the positions of the source and the receiver, $\boldsymbol{G}$ is the Green function for a homogeneous medium and $\boldsymbol{E}$ is the electric field for a homogeneous half-space. Equation (1) is an approximation for the low conductivity contrasts and it is very useful for an inversion, where $\boldsymbol{G}, \boldsymbol{E}$, and $\sigma_a$ are known and $\sigma(\boldsymbol{r})$ is unknown.

For the inversion we had to consider how the magnetic dipoles were used. We obtained the vertical and horizontal magnetic dipole (VMD and HMD, respectively) arrays as described by Pérez-Flores et al. (2012). The method we are using was developed and explained in Pérez-Flores et al. (2012). In the following paragraphs we will make a simple modification to the equations already published in order to accept arbitrary profile azimuths. The integral equation for VMD is

$$\sigma_{a,z}(\boldsymbol{r_1}, \boldsymbol{r_2}) \cong -\frac{16\pi s}{\omega \mu_0 m_z} \int_v \boldsymbol{G}_{H_z}(\boldsymbol{r}, \boldsymbol{r_2}) \bullet \boldsymbol{E}_{H_z}(\boldsymbol{r}, \boldsymbol{r_1}) \sigma(\boldsymbol{r}) dv \tag{2}$$

For HMD the integral equation in the $y$ direction is given by

$$\sigma_{a,y}(\boldsymbol{r_1}, \boldsymbol{r_2}) \cong -\frac{16\pi s}{\omega \mu_0 m_y} \int_v \boldsymbol{G}_{H_y}(\boldsymbol{r}, \boldsymbol{r_2}) . \boldsymbol{E}_{H_y}(\boldsymbol{r}, \boldsymbol{r_1}) \sigma(\boldsymbol{r}) dv \tag{3}$$

HMD in the $x$ direction is given by:

$$\sigma_{a,x}(\boldsymbol{r_1}, \boldsymbol{r_2}) \cong -\frac{16\pi s}{\omega \mu_0 m_x} \int_v \boldsymbol{G}_{H_x}(\boldsymbol{r}, \boldsymbol{r_2}) . \boldsymbol{E}_{H_x}(\boldsymbol{r}, \boldsymbol{r_1}) \sigma(\boldsymbol{r}) dv \tag{4}$$

The expressions for $\boldsymbol{G}_{H_z}, \boldsymbol{E}_{H_z}, \boldsymbol{G}_{H_y}, \boldsymbol{E}_{H_y}, \boldsymbol{G}_{H_x}$ and $\boldsymbol{E}_{H_x}$ can be consulted in Perez-Flores et al. (2012). VMD profiles can run at any angle (Eq. 2), but HMD profiles run only in either the *y-direction* (90⁰; Eq. 3) or *x-direction* (0⁰; Eq. 4). Arbitrary direction profiles like those observed around the Chac-Mool sinkhole (Fig. 3) constituted a problem. So, we had to modify Eq. (4 and 5) in order to accept the arbitrary angle profiles.

Using a simple rotation for $\boldsymbol{E}$ and $\boldsymbol{G}$ in terms of their vector components, the $y$- direction for HMD is

$$G_{H_y}(\boldsymbol{r},\boldsymbol{r}_2) = d\hat{\imath} + e\hat{\jmath}\,, E_{H_y}(\boldsymbol{r},\boldsymbol{r}_1) = \ a\hat{\imath} + b\hat{\jmath} \tag{5}$$

and the $x$-direction is

$$G_{H_x}(\boldsymbol{r},\boldsymbol{r}_2) = e\hat{\imath} + f\hat{\jmath}\,, E_{H_x}(\boldsymbol{r},\boldsymbol{r}_1) = \ b\hat{\imath} + c\hat{\jmath} \tag{6}$$

When we rotate Eq. (3) $90^0$, it becomes Eq. (4). So, we can find $\boldsymbol{E}$ and $\boldsymbol{G}$ in terms of their rotated components:

$$\begin{pmatrix} E_x \\ E_y \end{pmatrix} = \begin{pmatrix} cos\theta & sen\theta & 0 \\ 0 & cos\theta & sen\theta \end{pmatrix} \begin{pmatrix} a \\ b \\ c \end{pmatrix},$$

$$\begin{pmatrix} G_x \\ G_y \end{pmatrix} = \begin{pmatrix} cos\theta & sen\theta & 0 \\ 0 & cos\theta & sen\theta \end{pmatrix} \begin{pmatrix} d \\ e \\ f \end{pmatrix} \tag{7}$$

If an HMD profile runs at $0^0$, $(E_x,\ E_y)$ becomes $\boldsymbol{E}_{H_y}$ from Eq. (3). If the profile runs at $90^0$, $(E_x,\ E_y)$ becomes $\boldsymbol{E}_{H_x}$ from Eq. (4).

Thus, for an arbitrary angle profile, Eq. (3) and (4) become a single equation,

$$\sigma_a(\boldsymbol{r}_1,\boldsymbol{r}_2) = -\frac{16\pi s}{\omega\mu_0 m}\int[G_x(\boldsymbol{r},\boldsymbol{r}_2)E_x(\boldsymbol{r},\boldsymbol{r}_1) + G_y(\boldsymbol{r},\boldsymbol{r}_2)E_y(\boldsymbol{r},\boldsymbol{r}_1)]\sigma(\boldsymbol{r})dv \tag{8}$$

For terms $a$, $b$, $c$, $d$, $e$, and $f$ see Perez-Flores et al. (2012).

For the 3D inversion, we used Eq. (8) for the HMD profiles and Eq. (2) for the VMD profiles. We used 10, 20, and 40 m as the source–receiver separations for VMD and HMD in every profile. We pooled all data sets and performed a joint inversion to obtain a single 3D conductivity model. We simulated the heterogeneous half-space as a conglomerate of rectangular prisms. We assumed that conductivity in every single prism was constant, however unknown. Eq. (2) and (8) can be written as a linear equation system and in a matrix fashion:

$$\boldsymbol{\sigma}_a = \boldsymbol{W}\boldsymbol{\sigma} \tag{9}$$

where $\boldsymbol{\sigma}_a$ represents the column vector of apparent conductivities, matrix $\boldsymbol{W}$ contains the weights or products of the Green function and electric field and is partitioned for VMD and HMD, and $\boldsymbol{\sigma}$ represents the column vector of the real conductivities (unknowns). We used quadratic programing to minimize the following objective function, $\boldsymbol{U}$:

$$U(\boldsymbol{\sigma}) = \frac{1}{2}\left\|\boldsymbol{\sigma}_a - \mathbf{W}\boldsymbol{\sigma}\right\|^2 + \frac{1}{2}\beta\left\|\mathbf{D}\boldsymbol{\sigma}\right\|^2$$

$$\boldsymbol{\sigma}_{lower} < \boldsymbol{\sigma} < \boldsymbol{\sigma}_{upper} \tag{10}$$

Matrix $\boldsymbol{D}$ represents the first-order spatial derivatives of the contiguous prism conductivities. Parameter $\beta$ controls the smoothness of the 3D conductivity model; when it was low, we obtained a rough 3D model. The first term fits the apparent

conductivity data taken at the field. The second term in Eq. (10) contains the spatial derivatives of the conductivity in (x, y, z) direction. The smoothness parameter controls the magnitude of the second term. If zero, only the data was fit and the model use to be very rough; if very large, the model converged into a homogenous half-space. We transformed the Hessian to achieve diagonal unity. This way the smoothness parameter varies in a very narrow window. We tested the values 0.1, 0.01, and 0.001.

5    The 0.1 value yields a smooth model and the 0.001 value a rough model. We began with a smooth value that gave the simplest but the most probable model (according to the Occam's Razor principle), and we lowered the parameter to recover more structure; however, after a certain point the structure turned unreal from the geological point of view. The idea was to recover most of the structure while keeping the simplest and most probable model.

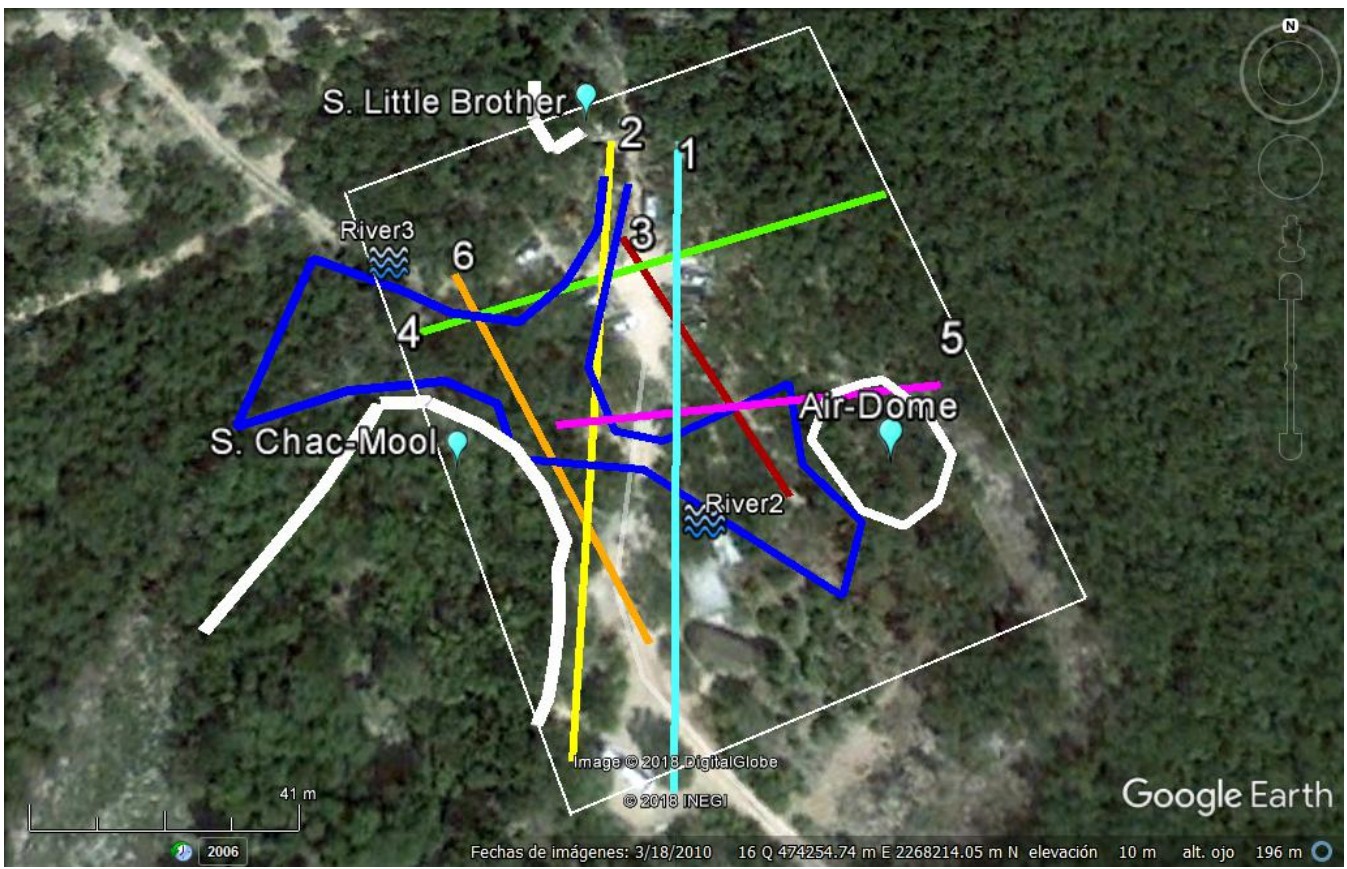

10    Figure 3. Profiles crossing underground water conduits in the sinkhole area (numbered lines). The white rectangle is the 3D modeled area. White lines mark the sinkhole boundaries. Dark blue lines are the suggested underground water conduits paths.

## 2. Resistivity cross-sections on the 3D model.

For the 3D inverse modeling we used an *(x, y, z)* grid of prisms, assuming constant conductivity in every prism. We performed the inverse modeling choosing *Δx=Δy =2.5* m in the *(x, y)*-directions because the EM measurements were taken every 5 m; the variable discretization of *Δz* was chosen to be (0, 2, 5, 8, 12, 18, 25, 35, and 50 m) and β=0.01 was the smoothness factor. Conductivity is the unknown, but we prefer to show the resistivity (the inverse of conductivity) results. In Figure 4 we present the 3D resistivity model after the inversion of whole sets of data. In that figure we present the interpolated resistivity cross-

sections under the six profiles. Blue indicates resistivity areas and red low resistivity. There are spaces between profiles with no data. The 3D model for those areas is not so reliable. Therefore, as a first approach, we show the model for the areas for which we had data. There is very good coherence where the model crosses. Figure 4 shows irregular paths for the two underground water conduits, according to the map from the divers (x, y, z). Water table depth in the open sinkholes is 7 m. The water conduits follow very intricate paths. We think that there are narrower river branches that have not yet been mapped

by the divers. Interestingly, some paths were marked below the resistivity areas. The upper water level of the subterranean river is probably far from the surface, making the rock mass or resistive mass, more structurally stable, or maybe those resistive bodies are air-filled caves over the water table. By *Resistive Mass (RM)*  we refer to the dry limestone rock between the surface and the ceiling of the cave and the air-filling the cave. We can idealize a typical cave in this area (near the coast), vertically consisting of dry limestone (resistive),  an air space (resistive), followed by fresh water (lower resistivity), the halocline (mixing

of fresh and salty water), and, at the bottom, salty water (lowest resistivity) surrounded by saturated limestones as bedrock. We cannot distinguish in the RM, what is dry limestone and what is air-filling the cave.

In Fig. 5 we show the six cross-sections obtained with the 3D resistivity model. Cross-section (a) corresponds to the profile-1 model, cross-section (b) to the profile-2 model, and so on. Every profile is indicated with a white circle, which pinpoint the

interpolated (x, y, z) hidden-water conduits. The (x, y, z) locations were obtained from the mapping made by scuba divers. We delineated the inferred cave section with a rectangle, because we could not see details. We assumed the saturated limestone was bedrock, because dry limestone resistivity was larger than 1000 ohms-m. In the 3D-model cross-sections, the bedrock looks green everywhere (160-170 ohms-m). Only some small sections were blue (1000 ohms-m).

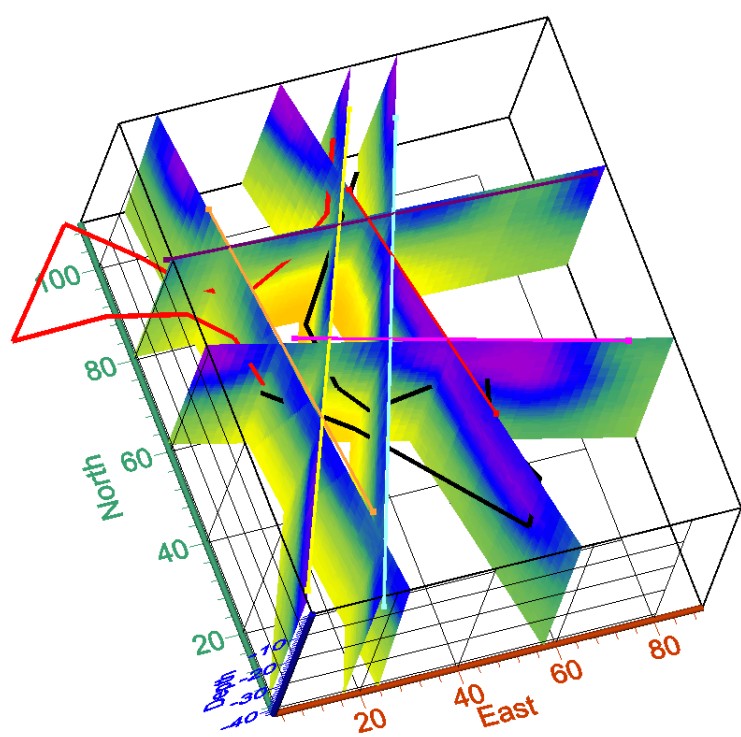

Figure 4. 3D resistivity model for the Chac-Mool sinkhole complex. Here, we show only the distribution of the cross-sections where the profiles were run. The red and black irregular lines represent the underground water conduits.

From the six resistivity cross-sections, we can see that most of the river crossings show a green color over them. This means that the subterranean water conduits are probably close to the surface and the thickness of the RM is therefore thin, meaning RMs in those areas are more vulnerable to sinking, though we did not find evidence of subduction or fracturing on the surface. The cross-section for profile 1 (Fig. 5a) shows three crosses: one at x=18 m showing a thin RM and the other two showing a thicker RM. Profile 2 (Fig. 5b) shows a green color, meaning thinner RMs. Profile 3 (Fig. 5c) shows one river crossing that is shallow and another deeper one. We clearly detected a shallower subterranean river (green color) using the EM-LIN equipment but it is not clear how much deeper it is. We must remember that the white circles are interpolations taken from the diver's map. The deeper river crossing coincides with the location of a large resistivity mass between zero and 20 m; this means that divers had to dive below this resistivity mass (1000 ohms-m). Profile 4 (Fig. 5d) shows three crossings with green color. Profile 5 (Fig. 5e) shows three crossings, two are deep (between z=20 m and z=30 m) and one is shallow (z=15 m). The deeper crossings are consistent with the reported diving depth and the thicker RM shown by the large resistivity mass. However, at x=25 m the river seems to be 10 m deeper, possibly because of the presence of a huge hard rock (very resistive). Profile 6 (Fig. 5f) shows a shallow river and a deeper one. Resistivities are consistent with the position of the river.

We know that divers swam throughout subterranean water conduits. In Fig. 5 we broadly suggest the location of the river crossings (rectangular polygon). Given the color descriptions in Fig. 5, we can say that blue is an indication of dry limestone RMs or dry limestone and air-filled caves at the top of the water conduit or close to the surface. The green color is so widespread that it surely indicates fresh water (50 to 70 ohms-m). Also, the resistivity cross-section shows a green color where the subterranean water conduits seem to be shallow. We expected to see a narrow blue coloration and green color over those shallow water conduits, but we did not, because the narrowest source–receiver separation at the EM34 was 10 m (too large to see surface details). In some way the estimated true conductivity is still an average. Maybe if we use a shorter separation, we could see a thinner blue color for the RM and then a green color for the fresh water. The transition from green to red (yellow) could be the transition from fresh water to salty water. We expect fresh water at the top and salty water at the bottom because of the density.

We drew the river section to emphasize that the resolution of the EM34 instrument is not good enough to sharply isolate the water conduits from the bedrock. A possible explanation is that the upper sections of unaltered bedrock (limestone) are partially saturated with fresh water (because of the 50-70 ohms-m values) and the deeper sections are saturated with salty water (because of the 6-10 ohms-m values). So, there are no large horizontal resistivity differences between the river location and the bedrock. It is almost certain that the permeability of the bedrock is as high as the permeability of the limestone at the surface. When it rains, water quickly disappears. Aerial-electromagnetics (flying 30 to 50 m over the surface) would yield an even lower resolution (Supper et al. 2009).

In profile 4 (Fig. 5d) there is a green color section close to x=70 m (red square). It is possible that there is a shallow subterranean river close to the surface that has not yet been mapped by the divers.

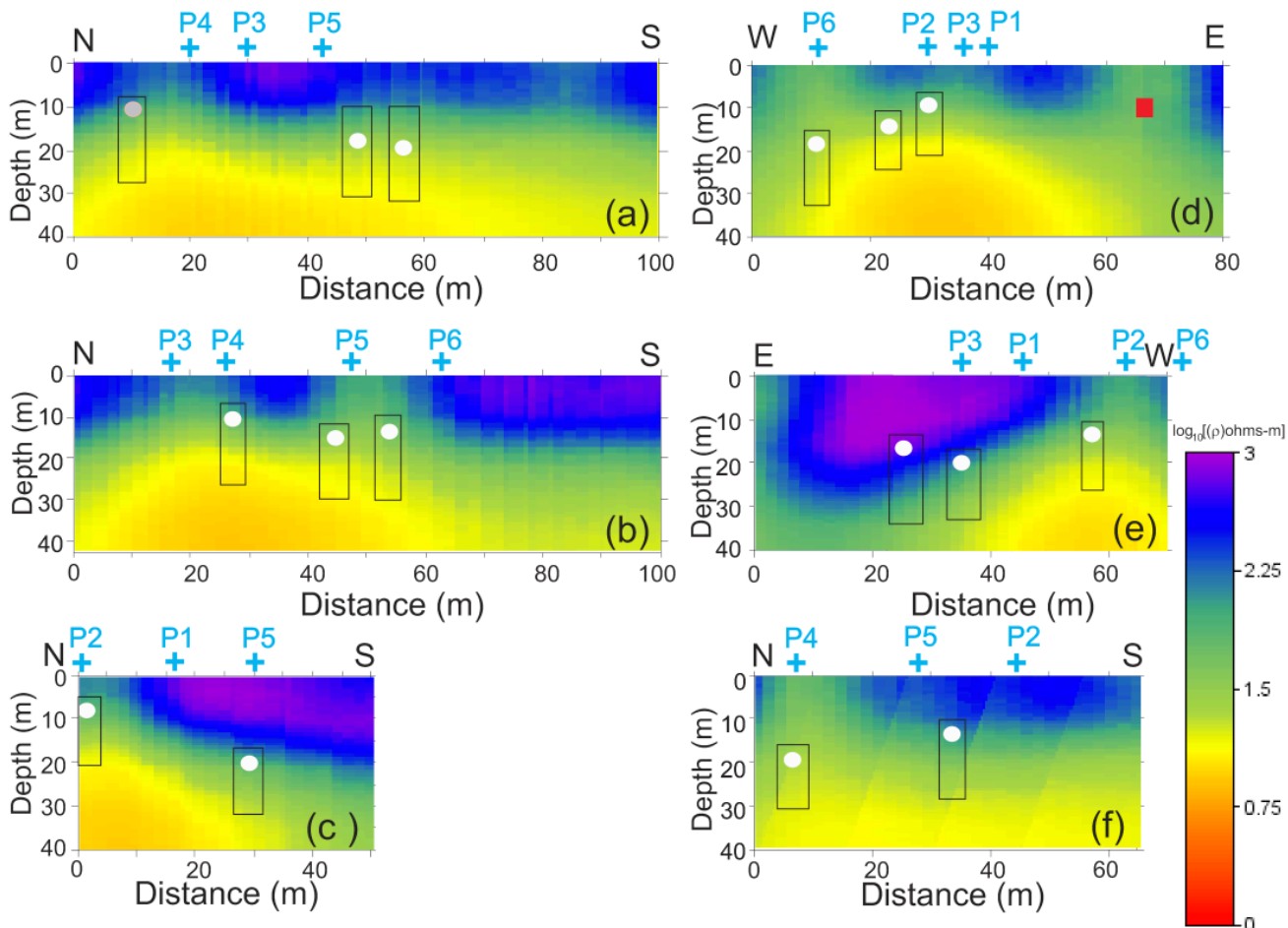

Figure 5. Cross-sections of the 3D resistivity model for profiles 1 to 6. Resistivity units are base 10 logarithm. Blue color indicates more resistive areas and red the least resistive areas. Blue numbers indicate the other profile crossings. White circles pinpoint the areas where scuba divers have mapped the underground water conduits. Red circles show the position of an underground river inferred from the model. The square polygon is a broad suggestion of the river tunnels.

## 2.1 Isometrics of the 3D resistivity model.

The Chac-Mool sinkhole system is a complex of three small sinkholes (Air-Dome, Little Brother, and Chac-Mool). According to divers, there are two underground water conduits. Their vertical variations may cause thinning of the limestone RMs and therefore sinking. According to the cross-section in Fig. 6, the EM-LIN equipment cannot sharply disitinguish between the subterranean river tunnels and the bedrock, maybe because there is not enough change in resitivity. This means that limestone bedrocks are partially saturated with water and therefore under a process of chemical disolution. The isometric view of the 3D

resistivity model (Fig. 6) shows the spatial distribution of the three sinkholes in the system, the two proposed water conduits and their paths and the location of the five EM-LIN data profiles.

The blue and orange surfaces are equal-resistivity surfaces in the 3D model. The blue surface shows the contact between dry limestone (resistive) and the fresh water (˜80 ohms-m). The resistive layer may contain unaltered limestone and/or air-filled

caves. It is very interesting that this layer outcrops where underground water conduits are very close to the surface, maybe because the shortest source-receiver distance (10 m) is larger than the RM thickness. This surface does not show where the sinkholes are, because of the lack of data. We did not manipulate the 3D model in order to force outcrops of areas with sinkholes. The orange surface represents the contact between the fresh water and the salty water (Halocline).

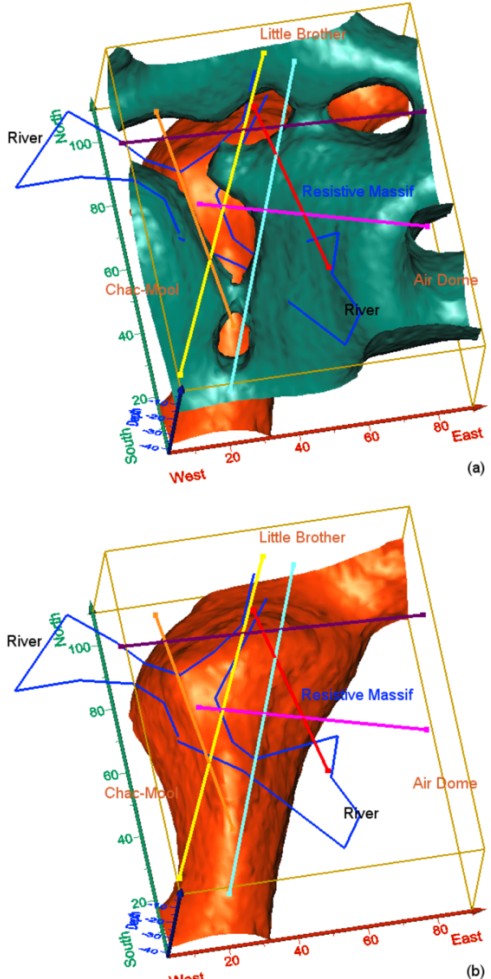

Figure 6. Isometric representation of the 3D resistivity model. Straight lines represent the EM profiles. (a) Blue iso-surface representing the bottom topography of the dry limestones. (b) Orange iso-surface representing the area where fresh and salty waters meet.

## 3. Conclusions

In this research we studied the Chac-Mool sinkhole complex by EM methods at LIN. These methods consist of a source loop and a receiver loop operating in two coplanar arrays VMD and HMD. These two arrays or polarizations view inside the Earth in two different ways. We used both arrays to perform a joint inversion and to obtain a single three-dimensional (3D) resistivity model. Equations had already been published for a mesh of perpendicular and parallel profiles but not for arbitrary angle profiles (Perez-Flores et al., 2012). In this research the profiles were taken inside the jungle and we took the advantage of man-made paths; however, these paths were located at arbitrary angles. We modified the existing equations and obtained a more general set of equations.

The 3D inversion of both VDM and HDM arrays led to a single 3D resistivity model. The cross-sections of this 3D model show the points where the underground water conduits cross. The areas where the underground water conduits are close to the surface could represent hazard zones because of the possibility of RMs collapsing. We also observed the distribution of fresh and salty waters and the areas where they meet or the transition surface (halocline). Our observations indicate that water conduits might run along tunnels, but the resistivity of those tunnels does not differ sharply from the resistivity of the bedrock, meaning that bedrock could be saturated with water (fresh and salty depending on depth). The isometric view shows that the resistivity iso-surface corresponds with the bottom topography of the underground RM. At the center of the area of study this RM seems to be very thick, indicating that this area is safe from sinking. This isometric view also shows the contact between fresh and salty water.

The EM-LIN technique is a fast, efficient, and inexpensive procedure for explorations over hard-rock sinkhole areas. It allows us to obtain the geometry of the underground water conduits and the distribution of fresh and salty water.

## 4. Discussion

EM-LIN methods can be applied for detecting natural caves under roads or caused by shallow mining or for determining resistivity changes caused by landsliding. This geophysical technique is ideal to detect resistivity changes underground and it has the advantage that no-electrode grounding is required, making it faster and cheaper than DC resistivity. The equations contained in this paper can be easily computed for the determination of 3D resistive/conductive bodies buried underground. The disadvantage is that commercial equipments were produced for few source-receiver separations limiting the resolution and the penetration depth. The equipment used for acquiring the data of this paper has no source-receiver separations lower than 10 m, which makes difficult to accurately resolve the very shallow thickness of the dry limestones.

We also point out that this EM method works better when the conductivity of the buried target is very different from the bedrock conductivity. In this paper we observed that the subterranean conduit conductivity was not very different from the bedrock conductivity as we expected. These results encourage us to make an accurate EM-LIN modeling in order to find an explanation of the low resistivity contrasts observed.

**Acknowledgements**

Many thanks go to CONACYT for the graduate scholarship. We also thank to CICESE for allowing us to use the geophysical equipment and CICY for enabling the facilities to run the research. Thanks to Fernando Herrera for his help in the field work.

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
