# Peer review of "3D Inverse modeling of EM-LIN data for the exploration of coastal sinkholes in Quintana Roo Mexico"

_Natural Hazards and Earth System Sciences, 2018_

## Referee Comment (RC1) · Anonymous Referee #1 · 8 Jul 2018

Dear editor and authors, I have reviewed manuscript NHESS_2018_180 submitted by Luis E. Ochoa-Tinajero et al., to Natural Hazards and Earth System Sciences and entittled "3D Inverse modeling of EM-LIN data to investigate coastal sinkholes in Quintana Roo Mexico". Manuscript deals about the application of 3d EM-Lin in order to evaluate near to the surface underground structure in the Chac-Mool sinkhole system at Mexico. Authors employ geophysics in order to evaluate preferential flowpaths in the underground and its potential relation with the karstified net in the subsoil by means geophysical data. Manuscript show interesting results but there are some subjects that require to be evaluated in the manuscript in order to allow data evaluation and interpretation by a reader of the manuscript. I include some subjects that can be of interest

to improve manuscript. • Authors indicate that there are previous woks in the area carried out by the Speleological Survey; It should be interesting to be able to compare the obtained data from geophysics and the available data from direct study. • It should be of interest to include a geological map in order to evaluate the geological characteristics from the area, its context but also, if available, hidrogeological information at the regional-local scale previous to the geophysical analysis. Moreover it can be also of interest to include a geomorphological map about the surficial indicators of karst activity and some photographs from the study area. This photographs will permit the evaluation of the survey conditions but also the karst characteristics from the study area. • There are not units in the representation from figure 4 (color scale), at figure 5 the scale color requires an inset (log 10 (r) means, r at logartimic scale but it lacks units and the figure caption requires to be rewritten as I am not sure that I am able to understand what is described. • Information related to the referenced small sinkholes from the area (2.1 chapter) requires to be included in the geological preliminary map but also in the geophysical models to be compared with the geophysical data. • Also at chapter 2.1 there is not an evaluation of the expected values for bedrock and the way to choose or discuss the origin of obtained values. If the analyzed units are rocks it can be difficult that they are complete saturated, that it is the explanation for such data. This requires to be more detailed interpreted and discussed. • At 2.1 authors describe how they interpret the presence of sinkholes in the area, however there is not reference to surficial-geomorphological data to be compare with or about the presence of sinkholes in the area to be compared with the geophysical data.. • What criteria has been used to select the 160 ohm/m for the separation of units in the geophysical model? • Do authors indicate that the "bottom topography of the limestone roof" but what they are referencing is "the topography of the limestone roof"? • About the interpretation and description, roof cannot be thick, this is a contact, then it is needed to correct "the roof appears to be very thick", or "the roof is very thin". After in the same paragraph authors indicate that the, what I interpret, the thickness of the level is thick, then the susceptibility to collapse is lower, does author have information

about the fracturation nets from the unit? Not necessarily from the local area, but the state of the massive can be evaluated in a regional scale to know if stability can be related to the fracturation state of the unit if authors want to evaluate collapse susceptibility or hazard. • At Figure 6. I suppose that this is a 3d view of the topography of the contact, but it is not clear to see it, Can authors include the isolines of topography, or two maps with the topography and by the other hand of the resistivity values?. In this sense, as previously pointed out, the selection of the resistivity values requires to be discussed in order to define if other values can be better to evaluate the 3D underground structure. • In order to evaluate data from the area, where the water level is expected to be? are there any change related to the water salinity in the geophysical data?.

---

## Referee Comment (RC2) · Anonymous Referee #2 · 21 Oct 2018

The paper is interesting but poorly structured. Authors have to devide the article as it accepted in Copernicus template

Introduction

Why you go to study sinkholes with EM method Overview literature in this field What principles of sinkhole or karst detection you are using denote goal of your research: methodology of 3D inversion, karst detection

Study area Geology, local conditions Topography

The Method EM34 device has to be describe: what parameter is measuredconductivity what units you are use (mmho?) How you convert it to resistivity? Penetration depth of every array (separation, VMD, HMD)

Data acquisition. What antennas (separation) you used during data acquisition What dipoles you used

Theory of data inversion

Results Show primary graphs of measurements along every line Analyse maps generated, sections

Discussion. It is important to discuss criterion of detection water saturation May be refer to similar survey You conclude on water saturate tunnels, but what conductivity or resistivity will be in unsaturated rock environment

References are written right apart from missprint in family of Bauer 2011 on p.2 Figures require legend

---

## Author Comment (AC1) · 31 Dec 2018

Dear reviewer.

Attending to your suggestions, we attach a zip file with the following documents:

1. ResponseReviewer: Response to reviewer. In black are the comments by the reviewer and in green are the answer from the authors. 2. Modified Red Green: Original manuscript with the modifications. In red is the eliminated part of the text and in green the modifications and corrections done by the authors. 3. Modified. Manuscript corrected by the authors without figures. 4. All_figures. All the figures with legends of the

manuscript.

Thanks in advance and best regards.

Luis Ochoa

Please also note the supplement to this comment:
https://www.nat-hazards-earth-syst-sci-discuss.net/nhess-2018-180/nhess-2018-180-AC1-supplement.zip

---

## Author Response (AR1)

Reviewer 1 answers:

To improve manuscript. Authors indicate that there are previous woks in the area carried out by the Speleological Survey; It should be interesting to be able to compare the obtained data from geophysics and the available data from direct study.

We added some lines in the manuscript saying that The (x, y) locations was obtained from the scuba divers map. The depth (z) is inferred from our 3D resistivity model. We sign the inferred cave section as a rectangle, because we cannot see details. Where dot is red is because is not reported by the scuba divers map, but we see a similar pattern where a river crosses. This location is inferred.

The scuba diver map is for tourists and it was difficult to extract the information we needed, as you can see.

It should be of interest to include a geological map in order to evaluate the geological characteristics from the area, its context but also, if available, hidrogeological information at the regional-local scale previous to the geophysical analysis.

We added some words in the Study Area Chapter to say that limestones are everywhere and terrain is very flat.

Moreover it can be also of interest to include a geomorphological map about the surficial indicators of karst activity and some photographs from the study area. This photographs will permit the evaluation of the survey conditions but also the karst characteristics from the study area.

**There is not surface manifestation about fractures or subduction. As you can see in the pictures. The profiles were taken over the clear way and we did not see any surface subduction.**

There are not units in the representation from figure 4 (color scale), at figure 5 the scale color requires an inset (log 10 (r) means, r at logarithmic scale but it lacks units and the figure caption requires to be rewritten as I am not sure that I am able to understand what is described.

**We corrected this, putting units on the plots.**

Information related to the referenced small sink- holes from the area (2.1 chapter) requires to be included in the geological preliminary map but also in the geophysical models to be compared with the geophysical data.

Profiles data were taken over flat paths on the jungle as you can see in the pictures. There were not small features over the ways to suspect about a sinking (we added a line in the manuscript about this). We also added some lines to the manuscript to emphasize that Geology is cretaceous limestones everywhere.

Also at chapter 2.1 there is not an evaluation of the expected values for bedrock and the way to choose or discuss the origin of obtained values. If the analyzed units are rocks it can be difficult that they are complete saturated, that it is the explanation for such data. This requires to be more detailed interpreted and discussed.

Here we show you a resistivity section very close to the chac-mool area. This was obtained by DC resistivity inversion of Dipolo-dipole, Schlumberger and Wenner data sets joint inversion in order to get a single resistivity model. Here we used a source-receiver separation of 5m. With this separation was possible to see the dry limestones close to the surface that we call it as *roof*. In x=60m there is a small sinking, meaning that a subterranean river is close and that collapse is possible. However, you cannot distinguish the resistivity change between the subterranean river and the bedrock. Even that here, we used a shorter source-receiver separation. We only see a green color disruption on the dry limestones and a disruption on the red color long body. We can not explain this since the geophysical point of view. Only salt water and shales can low the resistivity in the bedrock. That is why, we think that bedrock is in some way saturated of salty water that lows the resistivity. We have no other explanation. If you have one explanation, we will be very grateful.

In the EM inversion we do not recover the roof thickness sharply because the shortest sourcereceiver separation (10 m) was to large. The EM34 equipment has only separations of 10, 20 and 40 m.

---

## Editor Decision (ED1)

[revised manuscript text omitted]

---

## Author Response (AR2)

Comments to the reviewer #1

Abstract describes the eventual problematic in the study area about the hidrogeological characteristics of the Yucatán peninsula, however, the manuscript does not evaluate this problematic and has as objective a geophysical method that can give information about, but it is not explained how this information can solve the described problematic. This means that the main objective of the manuscript and description included in the abstract is not the own objective of the manuscript itself. The obtained information in the manuscript can collaborate, with the joint analysis of other techniques and methodologies, at the evaluation of the contamination of aquifers in the area, however this subject is not developed in the manuscript. In this sense, abstract does not describe what the manuscript evaluate, and differs from the obtained conclusions obtained in the manuscript, that it is centered in the geophysical data.

Yes, the main objective is the application of a non-usual geophysical tools to solve the sinkhole geometry. If the sinkhole would be the main objective we would rather apply many geophysical techniques as you mentioned. This is an electromagnetic tool that has been used few times for sinkholes and it has used mainly in the aerial way. That is why, we mentioned in the text that aerial EM has even worst resolution (they use 1D inversion and they are far from surface). We found just one paper about EM-LIN application for sinkholes on land, but the authors just put the raw data without any further processing (it is referenced). McNeill popularized this kind of equipment and he developed the 1D inversion. But geology requires 3D inversion for complex problems. We developed the 3D inversion method and it was published in Geophysics in 2012 for a public of mainly geophysicists (it is referenced in the text). In this manuscript, we are trying to popularize this EM tool and the 3D inversion method for non-geophysicists with an application to sinkholes. Along this research, we found that some caves were close to the surface, becoming this, more important by the risk of collapse. We do not know about rock mechanics and how to evaluate such a risk. As geophysicists we see that those caves are close to surface.

If you do not mind, we prefer to keep as the main objective, the application of this geophysical tool for a sinkhole problem.

We did some changes in the abstract. If you do not like, please tell me. My email is mperez@cicese.mx (Marco A. Perez-Flores). I would like to have a more interactive discussion. In such a way I can attend your suggestions faster.

There are sentences that require rephrasing, eg. "In order to guarantee the sustainable use of this groundwater resource knowledge on the hydrogeological characteristics, such as geometry and position, of caverns and sinkholes and the depth of the freshwater/saltwater mixing zone (halocline) is needed."; or "These authors performed EM-34 measurements but they did no further processing, like perform a geophysical inversion."

We saw the mistakes and we corrected them on the text. Thanks!!. The complete manuscript was also corrected by an Amglo-speaker.

Authors indicate that they do not show the results from the 2D inversion, something that can be of interest when a new processing approach is presented. However, later when authors explain their 3D approach, they are exhibiting 2D sections (or its integration that can be referred as a 2.5D distribution, see figs. 4 and 5 for example). In this sense, I am not sure about if the 3d analysis described represents a 2D analysis with a perspective view. In data inversion it should be of interest to include all data, it could be that the carried out inversion makes reference to the 2D inversion, because if a real 3D exists, they should present the map view of different depths, or geophysical bodies in 3D fashion.

In this manuscript we are not developing a new geophysical technic or new 3D inversion method. The new 3D inversion method was proved and published in the Geophysics Journal in 2012 (referenced in the text). As a short history, We began by applying 2D inversion and we saw many unreal small features in the 2D conductivity models that we did not like. This happen because data has 3D information and we were using a 2D inversion tool. We decided to do 3D inversion with the equation published in the paper from 2012, but we found that our profiles were done with arbitrary azimuth. So we have to spend some time to solve the problem by mean of a coordinate rotation of the original equations. In such a way, we got a unique equation valid for any angle. It is not a new 3D inversion method, it is the same. We will attach the 2012 paper, where you will see that such equations are there. And that the method was proved with the response of a known underground conductivity model and then recovered it as you said. In this manuscript, it is easy to proof that for azimuths of $O^0$ or $90^0$ degrees to this single equation, you will get the two equations already published in 2012. I explain this manuscript, but

we also did some modifications in text in order to be more clear. In the 2012 paper explained that the quality of the integral can be monitored. Such integral must be unity or one. With this coordinate rotated equation, the integral was already unity as it must be. We think it is not necessary to mention this in the manuscript, because we need to add at least two equations more and more text. But if you consider that it is import. We can do it.

Researchers from other universities are beginning to use these equations (Geophysics, 2012) and this generalized equation will be well accepted by them and others, we hope.

The 3D conductivity model obtained did not show those small features obtained in 2D, meaning that 2D inversion can not lead with the 3D geology complexity. The field data are truly 3D and if we use a 2D inversion tool, the software will show non-real 2D features in order to explain the fully 3D data.

Geophysicists use 1D and 2D inversion when the data is not enough for a 3D inversion, but also when the 3D inversion has not been developed. This is not the case. We did not want to show the 2D conductivity models because we would have to explain a lot, why the 2D and the 3D looks different even that they are very similar. Confusing the readers maybe.

In this manuscript we show cross-sections of the 3D conductivity model. We have found that it is very difficult to show static pictures of any 3D image to somebody. You need to use an interactive software to freely rotate the 3D image or a short video, if not, we could confuse more the reader.

If we put the 2D models in figure 4, we would see discrepancies where the models cross each other. Instead, with the cross-sections of the 3D model, you will not see any discrepancy in such crosses. The reader could be more confident, we think.

We propose to erase any word about 2D inversion in the manuscript. This is a 3D conductivity model only.

In the case of figure 5, I do not find a significative signal that can be related to the underground flow in the area or related to cavities in the underground, in this sense a higher detail and resolution of figures is required in order to evaluate the real meaning of these changes and the potential availability to identify them from the rest of the area (this can be done in a discussion chapter in the manuscript).

Yes, the resolution is not enough or what we would like. This equipment EM-34 uses only 3 separations (source and receiver) those are 10, 20 and 40 m. This is because for every separation the frequency must be different. But we also used VMD and HMD. VMD with 10 m should observe the shallowest targets. That is why we think that some water conduits are shallower than 10 m, but we do not know, the number in meters. HMD with 10 m looks deeper than HMD. We can not improve the resolution.

About other terminological issues, I am not sure if the term "underground rivers" is the most recommended due the characteristics from the area, I mean, that the use of underground rivers can mean many things. Here I suppose that this term make reference to a preferential water flow path through the carbonates by grouts, fractures and caves. This means that there could exist an underground conduit or group of conduits where water flow, more than an underground river itself.

We changed 'underground rivers' by 'underground water conduits. Is it Ok? Sorry, we are geophysicists.

In summary (mainly referring to the conclusions chapter) one of the main highlights from the article is to apply a previous mathematical method for the inversion of 2D profiles, in this case, with arbitrary directions. This is carried out on contrary than the usual parallel and perpendicular profiles net survey. The idea to perform the inversion by homogeneous directions is to simplify the inversion method and to evaluate, in a systematical pattern, the pixel size for the inversion (element or objet defined for the model). I am not sure if the only change in the orientation of the profiles can be performed by the orientation change of the data coordinates In this case what happens is that the data distribution for the inversion does not represent a homogeneous map view distribution, producing that this requires to be analyzed in detail in the discussion chapter.

It is only a 3D conductivity model not 2D. It is usual to do a mesh of many paralleled profiles in order to get a 3D model. Spaces between profile and profile it is so short, that the interpolation of the 3D model it is quite reliable. In this case, because of the jungle, we used the available pathways. In this case, the spaces between one profile and another could be large and the interpolation of the 3D model could be not very confident. We ran a 3D inversion because of the geology complexity, but we did not want to show a resistivity map at different depths, because there are large gaps without information. We are showing the 3D model exactly bellow the profiles. That is more reliable.

I believe that if we want to perform a new approach for data inversion modifying previous usual approaches, it requires comparing, for example, the results of the inversion by means usual and new data distribution. In this case, it has been applied a new mathematical approach without comparing if this results can be of application for the studied case. In this sense, the most recommended approach should be to perform the survey in two different survey grids and to compare results. In the case that the study area does not permit this approach, the mathematical simulation can be an alternative, at least, to evaluate if the approach can be of application in the area. The conclusions, where a integration of data and its application should be included, as stated in the abstract, is not evaluated or developed.

In Perez-Flores et al. (2012), we developed the 3D conductivity inversion and it was tested as you mentioned before. Here, we are using it. But we only did a simple modification to the equations for Horizontal Magnetic Dipoles (HMD). The integral is still unity or very close. That is the quality control parameter. For VMD the equations remain the same.

There it not a discussion chapter, moreover when geophysical results are ambiguous in terms of changes of the properties of the indirect obtained results, and moreover the direct data from the area does not present a clear correspondence with the models, the lack of discussion decrease the interest of the manuscript. In this sense, I am not sure if the geophysical model represent a good approach for the carried out analysis, at least looking at the presented results, when at the same time a change in the processing methodology is presented without to compare with previous methodologies, the results are not unambiguous, the comparison with the known data are not straightforward (or they are not interpreted and discussed in terms of resolution and accuracy).

We cannot improve the resolution as we explained before (VMD with 10 separation looks the shallowest targets). The method is not new in this manuscript, but we are presenting a generalized equation for VMD with profiles at any azimuth. The methodology is correct, we think.

Maybe the interpretation that we are obtaining from the resistivity images are not the better. We have been discussing between us, what more or what less to say. We tried several grids for the 3D model. This is the best. We spent several months running the programs varying the grid, the smoothing, etc. This is the best model that we got.

Attending to the document of answers to previous review, there are some subjects highlighted in the previous review, that has been answered but not included in the manuscript or the recommendations have not been considered.

The use of roof for the thickness of the carbonatic unit over the cavity induces to mistakes, as roof is a surface (later I will enter in this subject that was also highlighted in my previous review).

That is true, Roof is a surface. We are proposing RM (resistive mass). Because this RM is the sum of the dry limestone mass and the air-filling mass in the cave.

In that sense, when we said that RM is thinner than 10 m, means that maybe there is air-filling the cave and the dry limestone mass could be even thinner.

[Figure]

**Centro de Investigación Científica y de Educación Superior de Ensenada, Baja California**
**División de Ciencias de la Tierra**
**Departamento de Geofísica Aplicada**

Dear Natascha Topfer,

I want to move the authors order if possible. I want to switch the Perez-Flores position with my position Ochoa-Tinajero.

I am very busy attending other projects. Dr. Pérez-Flores is presently attending the correspondence with you and doing the changes requested by the reviewer. It is better that he continues as a first and correspondence author.

Thanks in advance.

M.Sc. Luis Ochoa-Tinajero
CICESE

Carretera Ensenada-Tijuana No. 3918, Fraccionamiento Zona Playitas
Ensenada, Baja California, México 22860
Tel.: (646) 175-0500, Ext. 26302, 26301; Fax: (646) 175-0567, http://www.cicese.mx

---

## Author Response (AR3)

Comments to Editor and Reviewer:

**Editor Decision: Publish subject to minor revisions (review by editor)** (18 May 2019) by Mario Parise
Comments to the Author:
Even though the paper has improved throughout the revised versions, it still does not completely respond to the referees' observations. In particular, I totally agree with the request by the referee to include a Discussion section. Authors are kindly invited to revise their manuscript, trying to fulfill all the referees' request

If I understood, you are asking me to add a Discussion section attending the suggestion made by the reviewer.

The follow letter contains the reviewer suggestions.
Dear editor and authors,
I have reviewed manuscript nhess-2018-180 and the author response letter included in the submission.
I want to thank the tone and detailed explanation from the revision letter.
In my previous review I make some comments regarding the inversion method, not for the inversion from parallel profiles, moreover from the integration of profiles with non parallel strikes. This was mentioned in the manuscript as a new approach. When comparing parallel profiles at constant distances between the measurement points, we can make the inversion more easily than profiles with different strikes and where distance between survey points is not constant. In this sense, the inversion requires to evaluate the anisotropic pixel for the integration that produces a new complexity in the inversion, and requiring the iteration and heterogeneous raw data for the analysis. This was one of the subjects that I pointed out in relation to the methodological processing routines that were applied in the manuscript.
Due to the description of some geological subjects referenced in the text, I propose to include some more geological information from the area, in order to define the contour conditions for the later geophysical model. Inverse data model produce potential multiple solutions, and from them the local geology can improve the election of the most probable solution. I am not making reference to the own inversion data model, I making reference to the evaluation of ranges for a cavity or change in water content to be interpreted as preferential flow-paths in the underground.
The resolution of the model is in general low, that is related to the intrinsic resolution from the used geophysical method and the low contrast of the looked for changes in the underground in the study area (this is not a critic, I am just trying to describe the context). This produce that if we include information from other sources (geological, geomorphological, surficial data) we can improve the model and to validate it more easily than without this kind of data.
I understand the problem of accessibility, the vegetation in the area, the eventual irregular topography and other handicaps, which defines the survey context and makes the obtained results and applied methodology a challenge to decipher the hidrogeological pathway in the area. This should be benefited by the data discussion in terms of data representativeness. Due to the complex problem, and the interest to find water in contexts such as described, one in the future can think that the proposed approach can be used to get data and solve the complex hidrogeological context; however each context can differ and produce that the results cannot be generalized. In this sense, the data discussion can permit to evaluate this data, but also to permit that anyone in the future that can try to reproduce the results, and get from the manuscript a complete description of the ambiguities that can be present.
The manuscript is a case-study that can be benefited with a try of generalization that can be obtained from a discussion in terms of resolution and effectiveness. This was the idea to include more geological data from the area.

I was confused because reviewer suggestions are two or three but they could take months to test the model as he said. But at the end of the letter I understood that he wanted a discussion

about the topics he pointed out. Finally, I understood that you both want a Discussion section including the topics mentioned by the reviewer.
Here is the Discussion section we added to the manuscript.

**4. Discussion**
This was the best resistivity model after many trials moving the 3D mesh and the smoothing parameter. The data collected in the field, put us in a challenge, because the profiles azimuth was arbitrary, forcing us to adapt equations but also in terms of resolutions in areas without data. In case of parallel profiles there are short gaps without data, but with arbitrary azimuth profiles, some data gaps are larger. This 3D resistivity model is confident close to the profiles, that is why, we decided to show mainly the 3D model as cross-sections along the original profiles.
We did not add constraints in the inversion but our models matched with the information given by the scuba divers. In cases where geology is complex and data is sparse, any kind of constraint can improve the solution thanks to quadratic programming.
Our 3D resistivity model could be better if the resistivity contrast between water conduits and bedrock would be larger.